# Overexpression of miR-125b in Osteoblasts Improves Age-Related Changes in Bone Mass and Quality through Suppression of Osteoclast Formation

**DOI:** 10.3390/ijms22136745

**Published:** 2021-06-23

**Authors:** Shota Ito, Tomoko Minamizaki, Shohei Kohno, Yusuke Sotomaru, Yoshiaki Kitaura, Shinsuke Ohba, Toshie Sugiyama, Jane E. Aubin, Kotaro Tanimoto, Yuji Yoshiko

**Affiliations:** 1Department of Calcified Tissue Biology, Hiroshima University Graduate School of Biomedical and Health Sciences, Minami-ku, Hiroshima 734-8553, Japan; shota0313@hiroshima-u.ac.jp (S.I.); tatsu3@hiroshima-u.ac.jp (T.M.); kohnos@hiroshima-u.ac.jp (S.K.); 2Department of Orthodontics and Craniofacial Developmental Biology, Hiroshima University Graduate School of Biomedical and Health Sciences, Minami-ku, Hiroshima 734-8553, Japan; tkotaro@hiroshima-u.ac.jp; 3Natural Science Center for Basic Research and Development, Hiroshima University, Minami-ku, Hiroshima 734-8551, Japan; sotomaru@hiroshima-u.ac.jp; 4Department of Bioengineering, The University of Tokyo Graduate School of Medicine, Bunkyo-ku, Tokyo 113-0033, Japan; yoshiakikitaura940@gmail.com; 5Department of Sensory and Motor System Medicine, The University of Tokyo Graduate School of Medicine, Bunkyo-ku, Tokyo 113-0033, Japan; s-ohba@nagasaki-u.ac.jp; 6Department of Animal Science, Niigata University Graduate School of Science and Technology, Nishi-ku, Niigata 950-2181, Japan; sugiyama@agr.niigata-u.ac.jp; 7Department of Molecular Genetics, University of Toronto, 1 King’s College Circle, Medical Sciences Building, Toronto, ON M5S 1A8, Canada; jane.aubin@utoronto.ca

**Keywords:** miR-125b, matrix vesicles, bone development and aging, fracture healing

## Abstract

We recently reported an unexpected role of osteoblast-derived matrix vesicles in the delivery of microRNAs to bone matrix. Of such microRNAs, we found that miR-125b inhibited osteoclast formation by targeting *Prdm1* encoding a transcriptional repressor of anti-osteoclastogenesis factors. Transgenic (Tg) mice overexpressing miR-125b in osteoblasts by using human osteocalcin promoter grow normally but exhibit high trabecular bone mass. We have now further investigated the effects of osteoblast-mediated miR-125b overexpression on skeletal morphogenesis and remodeling during development, aging and in a situation of skeletal repair, i.e., fracture healing. There were no significant differences in the growth plate, primary spongiosa or lateral (periosteal) bone formation and mineral apposition rate between Tg and wild-type (WT) mice during early bone development. However, osteoclast number and medial (endosteal) bone resorption were less in Tg compared to WT mice, concomitant with increased trabecular bone mass. Tg mice were less susceptible to age-dependent changes in bone mass, phosphate/amide I ratio and mechanical strength. In a femoral fracture model, callus formation progressed similarly in Tg and WT mice, but callus resorption was delayed, reflecting the decreased osteoclast numbers associated with the Tg callus. These results indicate that the decreased osteoclastogenesis mediated by miR-125b overexpression in osteoblasts leads to increased bone mass and strength, while preserving bone formation and quality. They also suggest that, in spite of the fact that single miRNAs may target multiple genes, the miR-125b axis may be an attractive therapeutic target for bone loss in various age groups.

## 1. Introduction

MicroRNAs (miRNAs) are small non-coding RNAs, comprising about 20–25 nucleotides each, and are found in most organisms [1,2,3]. miRNAs control gene expression post-transcriptionally by binding to target mRNAs and thereby regulating their translation and stability [4]. The human genome encodes approximately 2000 miRNAs [5], many of which regulate physiological and pathological events in a variety of cell and tissue types [6]. Valadi et al. first observed that exosomes (one type of extracellular vesicle (EV)) contain miRNAs that can be delivered to recipient cells [7] and targeted to specific mRNAs [8]. For example, miR-214-3p in osteoclast-derived exosomes was transferred to osteoblasts, resulting in decreased osteoblast activity and bone formation in mice [9]. Matrix vesicles (MVs) budding from osteoblasts, chondrocytes, and odontoblasts appear to be specialized EVs that accumulate in unmineralized extracellular matrix in hard tissues to initiate hydroxyapatite crystal formation [10,11,12,13]. MVs from chondrocyte [14,15] and osteoblast [16] cultures contain miRNAs. Interestingly, the enrichment profiles of miRNAs in osteoblast and chondrocyte MVs are distinctly different, suggesting that their respective MVs may exhibit distinct roles in bone and cartilage.

Of miRNAs expressed in osteoblasts, we found that miR-125b is selectively transferred to bone matrix as MV cargo and released into the bone marrow microenvironment during bone resorption to downregulate osteoclastogenesis via the miR-125b-mediated inhibition of the expression of PRDM1, a transcriptional repressor of anti-osteoclastogenesis factors [16]. Consistent with this, we generated transgenic (Tg) mice overexpressing miR-125b under the control of the human osteocalcin (hOC) promoter. As expected with this promoter [17], we detected overexpression of miR-125b in osteoblasts but not osteoclasts or other organs tested in the Tg mice [16]. The Tg mice exhibited a marked increase in trabecular bone volume with a decreased number of osteoclasts. Interestingly, osteoblast development and bone formation appeared normal in this Tg model, and the mice grew normally at least up to 19 weeks of age [16], even though decreased bone resorption usually negatively impacts skeletal development and aging in humans and mice [18,19,20]. The latter is the case, for example, in osteopetrosis, where deficiencies in osteoclast development and/or activity cause skeletal anomalies including short stature in mice and humans [18]. Not only *Tnrsf11a*-knockout mice but also *Nfkb1/2*-deficient mice develop osteopetrosis-like symptoms such as metaphyseal flaring, shortened long bones, disorganization of chondrocytes at the growth plate and/or almost complete occlusion of the marrow cavity [19,20]. Such observations prompted us to further investigate the effects of osteoblast-mediated miR-125b overexpression on skeletal morphogenesis and remodeling both during development and in aged mice. Because antiresorptive agents can have deleterious effects on fracture healing [21], we also used a femoral fracture model to assess potential effects of the MV-miR-125b axis in the process.

## 2. Results

### 2.1. Aberrant Bone Remodeling Is Detectable during the Peripartum Period in Tg Mice Overexpressing miR-125b in Osteoblasts

To determine whether and how overexpression of miR-125b in osteoblasts affects developing bone, we surveyed the structural features of femurs and/or tibiae from E16.5 to P5 wild-type (WT) and Tg mice (longitudinal sections in Figure 1A,B). Through the peripartum period analyzed, we found no differences in cartilage or trabecular architecture in the primary spongiosa of Tg versus WT mice. However, Tg mice exhibited more trabecular bone in the mid-diaphysis than WT mice, and abundant Alcian blue-staining cartilage matrix was seen within the Tg trabeculae as early as E19.5 (Figure 1B). Double calcein-Alizarin red S labeling of the mineralization front in femoral cross-sections taken from the mid-diaphysis revealed no detectable differences in bone development on the periosteal side, and bone collars became thicker in an age-dependent manner in both genotypes (Figure 2A,B). Notably, however, calcein-labeled bone had mostly disappeared in WT but not Tg mice at P2 (Figure 2A,C), paralleling a larger WT compared to Tg marrow cavity (Figure 2B, see also Figure 2G). Concomitant with these observations, the number of TRAP-positive (TRAP^+^) multinuclear cells (MNCs) per calcein-labeled bone area was lower in Tg P2 mice than that in WT P2 mice (Figure 2D,E). Micro-computed tomography (CT) of tibiae revealed that bone morphometric parameters were not significantly different between genotypes up to P2 (Appendix A; Figure 2F,G). However, at P5, Tg mice exhibited higher bone volume/tissue volume (BV/TV) concomitant with a smaller marrow cavity compared to WT mice, without significant differences in the bone collar length between genotypes (Figure 2G). Taken together, these findings suggest that bone formation during early skeletal development in Tg mice is equal to that in WT mice, but that bone resorption is delayed in Tg compared to WT mice (see the schematic model in Figure 2H, based on [22]).

### 2.2. Age-Dependent Changes in Bone Parameters Are Different between WT and Tg Mice

To determine whether and how the early anomalies in bone parameters seen in miR-125b-overexpressing mice affected skeletal aging, we assessed not only bone but several other phenotypic traits in Tg and WT mice up to 77 weeks of age, i.e., a lifespan over which age-dependent bone loss would normally be manifest [23]. There was no difference in the average weight gain between Tg and WT mice from 5–77 weeks old (Figure 3A). Tg mice exhibited higher bone mineral density (BMD), BV/TV, trabecular thickness (Tb.Th), and trabeculr number (Tb.N) and lower trabeculr separation (Tb.Sp) than WT mice at three different ages assessed corresponding to young adult, middle-aged and old mice (10-, 30- and 77-week-old) (Figure 3B,C). All bone morphometric parameters (except Tb.Th) decreased or increased in mice of both genotypes by 77 weeks of age, but the decrease in BV/TV and Tb.N was less in Tg compared to WT mice (Figure 3B,C). At all ages, the bone marrow cavity was occupied by significantly more bone in Tg compared to WT mice (Figure 3B,D), leading us to assess whether hematologic abnormalities including anemia, which can occur in osteopetrosis-like conditions [24], are present. However, we found no hematologic anomalies in Tg mice (Table 1).

We used Fourier transform infrared (FT-IR) spectroscopy to assess the mineral (PO_4_^3–^)/matrix (amide I) content in bones of WT and Tg mice at 10 and 77 weeks old. While levels of both PO_4_^3–^ and amide I were comparable in WT and Tg bones at 10 weeks, PO_4_^3–^ but not amide I levels decreased in 77-week-old WT but not Tg trabecular bones (Figure 3E). The mineral/matrix ratio was not significantly different and did not change between 10 and 77 weeks old in the cortical bone of either strain, but the ratio in trabecular bone was significantly higher in Tg compared to WT mice at 77 weeks old (Figure 3F). A three-point bending test to assess bone strength and stiffness indicated that bone strength was highest in both 77-week-old WT and Tg mice, and that Tg bones were stronger than WT bones at both ages (Figure 3G). There was no difference in the stiffness of WT versus Tg bones or with age of either strain (Figure 3G). To address the limitation inherent in bending tests, further studies are warranted. However, taken together, the data suggest that Tg mice have a consistently higher bone mass and strength than WT mice but relatively few other marked bone anomalies, i.e., small mineral/matrix differences in the trabecular bone compartment only and no change in stiffness, as the mice age.

### 2.3. Fracture Healing Is Delayed in Tg Compared to WT Mice

Given the difference in bone parameters between WT and Tg mice with overexpression of miR-125b in osteoblasts, we used a femoral fracture model to determine whether Tg mice exhibit aberrant bone repair [25]. CT scans (Figure 4A) revealed time-dependent differences between the two genotypes in BMD and callus bone volume from 2 weeks to 8 weeks postoperation (Figure 4B). Calluses formed with similar timing and bone parameters were similar in the two genotypes early (2 weeks) post-fracture, but whereas WT calluses began to decrease by 6 weeks, Tg calluses, which were relatively larger than WT, did not, suggesting a delay in callus resorption in Tg mice. Histological observation (Figure 4C–F) confirmed that bone segments were connected with soft callus by 2 weeks postoperation in both genotypes, with cortical bone adjacent to the fracture partially resorbed in WT but not Tg mice. While cartilage and connective tissue were almost completely replaced by bone in WT mice at 6 weeks post-operation, the same was not seen in Tg mice, in which calluses at 6 weeks still had abundant cartilage (Figure 4D,E). Indeed, complete healing of the Tg fractures was not seen even at 8 weeks post-operation.

## 3. Discussion

We reported previously that osteoblasts express miR-125b that is selectively transferred to bone matrix as MV cargo and released into the bone marrow microenvironment during bone resorption to downregulate osteoclastogenesis via the miR-125b-mediated inhibition of the expression of the transcriptional repressor PRDM1 [16]. Tg mice overexpressing miR-125b under the control of the human osteocalcin promoter also exhibited downregulation of osteoclastogenesis leading to a marked increase in trabecular bone volume with apparently normal osteoblast development and bone formation, as well normal growth and weight up to 19 weeks of age. To further address the consequences of this apparent uncoupling of bone formation-bone resorption on skeletal development and the bone loss typically seen in the aging skeleton, we now report detailed analysis of the skeletons of Tg versus WT mice from E16.5 to 77 weeks of age, and in a situation requiring bone repair, i.e., fracture healing.

As expected, given that we used the osteocalcin promoter, which is not expressed until late mouse embryogenesis/shortly before birth in mature osteoblasts [17,26], development of the growth plate, primary sponsiosa and longitudinal bone growth in Tg tibiae in embryonic and neonatal stages appeared normal. However, the islands of cartilage frequently seen in bone trabeculae in the mid-diaphysis of the long bones of E19.5 and older Tg mice suggest a delay in the replacement of mineralized cartilage by bone due to a decline in osteoclastic bone resorption in the regions that exclude subchondral bone. This is consistent with our earlier observations in which we saw no reduction in numbers of TRAP^+^ MNCs in the calcified cartilage-subchondral bone zone, which may reflect lower accumulation of miR-125b in bone at the chondro-osseous junction in Tg mice [16]. In this regard, miR-125b has not been reported in lists of miRNAs in MVs budding from chondrocytes [14,15].

Analyses of cross-sections at the mid-epiphysis of femurs labeled with calcein and Alizarin red suggest that WT and Tg mice exhibit similar rates of intramembranous ossification. In contrast to WT mice, however, calcein-labeled bone was retained concomitant with a decreased number of osteoclasts on the calcein-labeled bone in P2 Tg mice. The fact that BV/TV was significantly decreased as expected in WT mice between P2 and P5 is consistent with the expected endosteal bone resorption resulting in thinner cortical bone and an expanding bone marrow cavity at this developmental age (see also Figure 2H) [22]. The increased BV/TV and smaller marrow cavity in Tg mice are consistent with their reduced number of osteoclasts and bone resorption. It is also worth noting that in spite of the copious trabecular bone within the bone marrow cavity, we detected no hematologic differences in peripheral blood between Tg and WT mice. However, further investigation is needed to explore the possibility of extramedullary hematopoiesis inTg mice [24].

It is well known that bone mass decreases with aging, with a resultant reduction in bone strength [23,27]. Bone quality is also critically involved in its strength [28,29,30]. In this regard, in at least some preclinical models of osteopetrosis due to defective osteoclast number and/or activity, the increased bone mass is associated with reduced bone mechanical strength [31]. It is also well established that a decrease in bone turnover due to osteoclast defects and reduced resorption reduces bone quality, with measurable changes in such bone parameters as Ca/P ratio, the crystallinity of bone mineral and collagen crosslinking [32,33]. Tg mice exhibited greater strength in bone than WT at the ages tested, consistent with their higher BMD and BV/TV (and other parameters, Figure 3). Tg mice also showed a smaller percent decrease in these parameters with age. In the mouse strain (C57BL/6J) used in this study, bone strength and stiffness as well as bone mass reach a maximum at about 50 weeks of age and gradually decline thereafter [29]. Thus, more work is required to characterize in detail the biochemical and structural properties of the Tg bone in older mice. Nevertheless, it is notable that the expression of osteoblast-associated genes and bone formation appear the same in Tg compared to WT mice [16], suggesting that it is the lower residual resorption in Tg mice that accounts not only for the preservation of bone mass but also the higher mineral/matrix ratio in and greater bone strength of Tg compared to WT mice at least up to 77 weeks of age.

The apparent preservation of bone mass and greater bone strength in older Tg compared to WT mice suggests that the miR-125b axis may be a useful therapeutic candidate for conditions affecting bone [16]. In this regard, however, the potential consequences of reduced bone resorption must be considered. For example, antiresorptive drugs appear to impair fracture healing [21], as has been documented with incadronate, which delays callus remodeling and fracture healing in a rat femoral fracture model [34]. Alendronate and denosumab also delay the removal of cartilage and the remodeling of the fracture callus, but strength and stiffness are enhanced when compared with control bones in a mouse femoral fracture model [35]. Our results comparing callus remodeling during femoral fracture healing over an 8-week period in Tg and WT mice are reminiscent of the results on fracture healing in these latter two studies. Thus, overexpressing miR-125b in osteoblasts under the control of the human osteocalcin promoter is presumed to affect the relative dominance of bone formation versus bone resorption not only during development and aging but also in a situation of skeletal repair, i.e., fracture healing.

## 4. Materials and Methods

### 4.1. Animals

C57BL/6J mice were obtained from CLEA (Tokyo, Japan). Tg mice overexpressing miR-125b in osteoblasts under the control of human osteocalcin promoter were developed on the C57BL/6J background [16]. Fetal and newborn mice (up to P5) were used regardless of sex, since this strain exhibits no differences in femoral bone parameters at least up to P7 [36]. Previously, we confirmed that male and female Tg mice exhibited the same skeletal phenotypes [16], but we used male mice for most (except for hematology) of the postnatal studies to avoid potential sex differences. Tg mice were identified by PCR using KOD FX Neo (Toyobo, Osaka, Japan) and primers for rabbit *Hbb* intron as follows: 5′-CTG GTC ATC ATC CTG CCT TT-3′ and 5′-TTA AGC TTA CAA AGA ATG GCC ACA GG-3′ [37].

### 4.2. Histology and Histomorphometry

Tibiae and femurs of mice were fixed in 4% paraformaldehyde (PFA) in PBS at 4 °C overnight, decalcified in 10% EDTA in PBS at 4 °C for 7–10 days and embedded in paraffin. To prepare plastic sections, fixed bones were embedded in glycol methacrylate (Technovit^®^ 7100; Kulzer, Hanau, Germany) or methyl methacrylate, monomer (Osteoresin Embedding Kit; FUJIFILM Wako Pure Chemical Corporation, Osaka, Japan). Paraffin sections (4 μm in thickness) were subjected to hematoxylin and eosin, Alcian blue (pH2.5), and tartrate-resistant acid phosphatase (TRAP) staining [38]. To estimate calcification, von Kossa staining was performed [39].

Calcein (5 mg/kg) and Alizarin red S (25 mg/kg) were injected intraperitoneally into pregnant mice at embryonic day 14.5 (E14.5) and E15.5, respectively. Male and female fetuses were collected at E16.5, E17.5, E18.5 and E19.5, and histmorphometry was performed at the mid-diaphysis in femoral plastic sections (4 μm in thickness) under fluorescence microscopy (Leica, Wetzlar, Germany). ImageJ (open source software developed by Wayne Rasband) was used to measure labeling areas.

### 4.3. μCT Analysis

μCT imaging of bones dissected from mice was obtained (Skyscan 1176; Bruker, Billerica, MA, USA) under the following conditions: 50 kV, 0.5 mA, 0.5 mm or 0.2 mm aluminium filter with a voxel size of 8.75 or 17.5 μm. Three-dimensional reconstruction was performed using NRecon (Bruker). Regions of interest in tibiae and peripartum femurs were set at 7 mm in width 0.5 mm below the growth plate and 0.5 (E16.5–19.5) or 1 mm (postnatal day 2 (P2) and P5) in width at midshaft, respectively. BMD, BV, TV, BV/TV, Tb.Th, Tb.N, Tb.Sp were calculated.

### 4.4. FT-IR Spectroscopy

Tibiae of male mice were fixed in 70% EtOH and embedded in Osteoresin, and sections (4 μm in thickness) were prepared on BaF_2_ slides (Pier-Optics, Tatebayashi, Japan). FT-IR spectra were obtained using the Spectrum 100 Optica (PerkinElmer, Waltham, MA, USA) and Spotlight 400 (PerkinElmer, Waltham, MA, USA). The microscope was equipped with a computer-controlled x/y stage that permitted spectral sampling of the tissue in defined steps within a rectangular area. IR spectra were collected with an aperture diameter of 50 μm, and transmission from 4000 to 700 cm^–1^ with a spectral resolution of 4 cm^–1^ using an MCT detector. One hundred and twenty-eight scans per point were collected and averaged.

### 4.5. Three-Point Bending Test

Bone biomechanical strength was assessed on fixed femurs of male mice using a CR-500-DX (Sun Scientific, Tokyo, Japan). Femurs were placed on the holding device with supports 4 mm apart, the upper loading device was aligned to the center of the femoral shaft on the anterior side, and mechanical load was applied at 1 mm/min until fracture occurred. Maximum load (N) and stiffness (N/mm) were calculated using the load–displacement curve.

### 4.6. Femoral Fracture Model

Femoral shaft fractures were made as described previously [40]. Briefly, under anesthesia (2.0% isoflurane), a 10 mm long incision was made along the femoral shaft of Tg and WT male mice (10 weeks old), and muscles were pushed aside to expose the left femur at the lateral side. An incision was made over the right knee joint, and the distal end of the femur was perforated, followed by the insertion of a 23-gauge spinal needle into the bone marrow cavity to make the guide hole. A transverse osteotomy in the midshaft region of the femur was performed using a diamond disc cutter with pouring water. A 23-guage spinal needle was inserted into the guide hole as the bone anchoring nail (Appendix A), and skin incisions were sutured. The success of bone repositioning was confirmed by μCT (Appendix A). The fractured femurs were then examined at 2, 4, 6, 8 weeks after surgery.

### 4.7. Hematology Test

Blood was collected from 10-week-old female mice and mixed with 1.0 mg/mL EDTA-2Na. After centrifugation, plasma was subjected to measure WBC, RBC, hemoglobin, mean corpuscular volume, mean corpuscular hemoglobin, mean corpuscular hemoglobin concentration, and platelet (FUJIFILM Monolith, Tokyo, Japan).

### 4.8. Statistical Analysis

Data are expressed as mean ± S.D. Statistical analysis was performed using JMP12 (SAS institute, NC). Comparisons between multiple groups and two groups were performed using Tukey’s multiple comparison method and Welch’s *t* test, respectively.

## 5. Conclusions

Early bone formation and mineral apposition rate in Tg mice overexpressing miR-125b in osteoblasts by using human osteocalcin promoter were comparable with those in WT mice. However, Tg mice exhibited a delay in medial (endosteal) bone resorption with a reduction in osteoclast numbers and hence increased trabecular bone. Age-dependent changes in bone mass, composition and mechanical strength were less effective in Tg than WT mice. Concomitant with these changes, miR-125b overexpression in osteoblasts impinged on calls resorption but not formation in a femoral fracture model. Thus, modulating the miR-125b axis in osteoblast–osteoclast communication may provide clinical benefits.

## Figures and Tables

**Figure 1 ijms-22-06745-f001:**
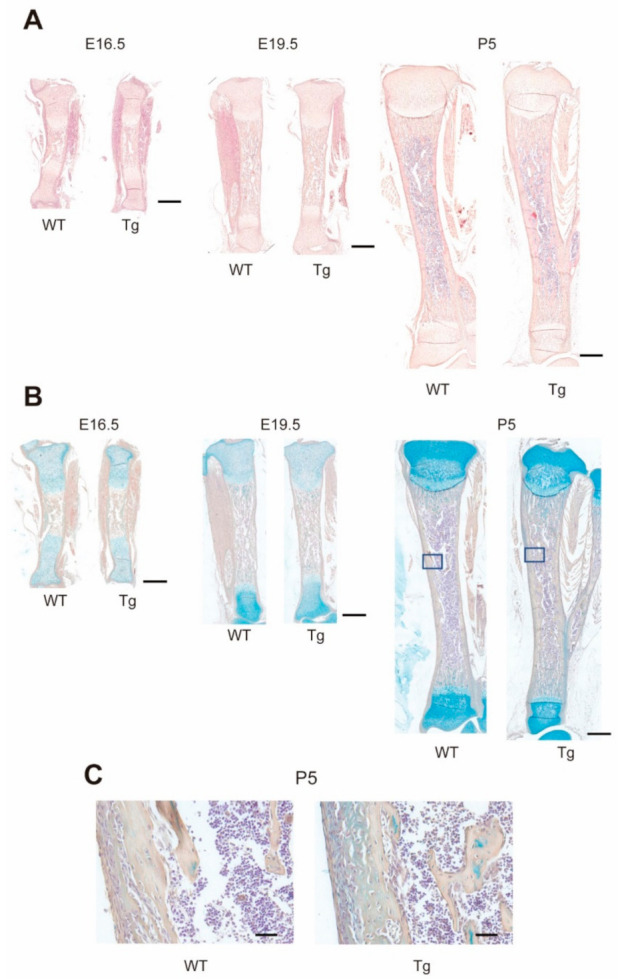
Longitudinal growth of Tg and WT tibiae during the peripartum period. (**A**,**B**) Representative images of longitudinal sections of tibiae at E16.5–P5, stained with hematoxylin-eosin (H-E) (**A**) and hematoxylin and Alcian blue (**B**). Scale bars, 500 μm. (**C**) High magnification images of the boxed areas (P5) in panel (**B**). Scale bars, 80 μm.

**Figure 2 ijms-22-06745-f002:**
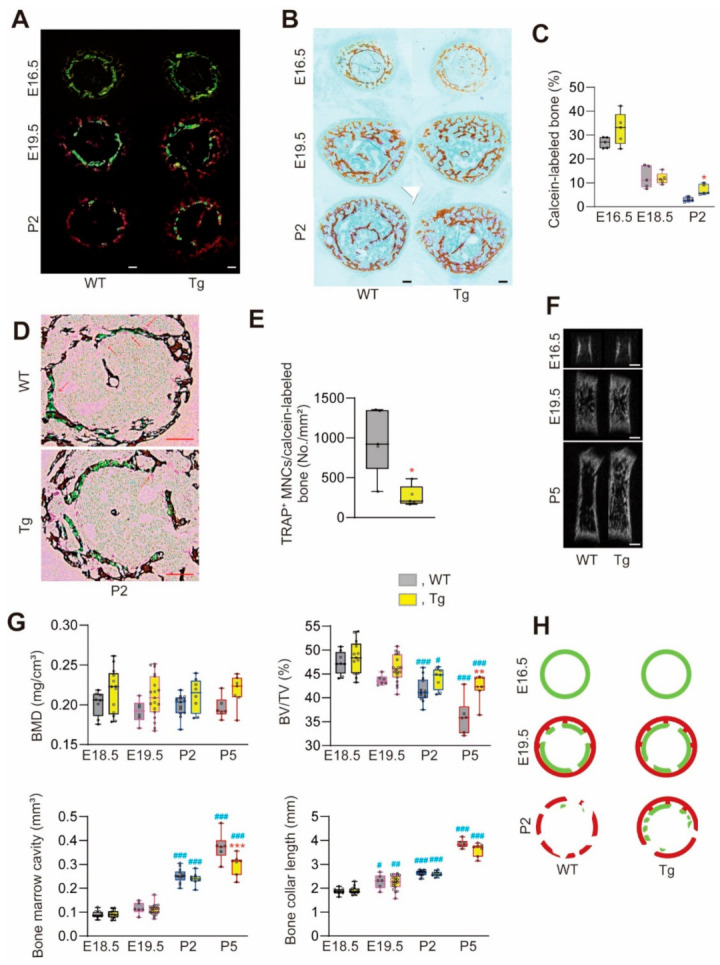
Lateral growth of Tg and WT femurs during the peripartum period. Calcein and Alizarin red S were injected into pregnant mice at E14.5 and E15.5, respectively. (**A**,**B**,**D**) Representative images of cross-sections of femurs at E16.5, E19.5 and P2. (**A**) Calcein and Alizarin red S-labeled bones. Scale bars, 100 μm. (**B**) Von Kossa staining of **A**. (**C**) The ratio of calcein^+^ to von Kossa^+^ areas analyzed by ImageJ. n = 5. * *p* < 0.05 versus matched WT. (**D**,**E**) Representative images of TRAP (purple) and von Kossa (brown) staining merged with calcein staining and the number of TRAP^+^ MNCs at the surface of calcein-labeled bones (both endosteal and periosteal side), respectively. Sections at P2 were analyzed, and methyl green was used as counterstain. Arrows, TRAP^+^ MNCs. Scale bars, 100 μm. n = 5. * *p* < 0.05. (**F**) Representative μCT images (longitudinal axis) at E16.5, E19.5 and P5. Scale bars, 500 μm. (**G**) μCT parameters in femurs of mice at E18.5–P5. ROI, 0.5 mm and 1 mm thickness at the midshaft of femurs of mice at E18.5–19.5 and P2–5, respectively. BMD, bone mineral density; BV/TV, bone volume/tissue volume; bone marrow cavity shown as TV excluding BV; bone collar length shown as the length from mesial to distal end of bone collar. n = 6–19. ** *p* < 0.01, *** *p* < 0.001 versus matched WT; ^#^ *p* < 0.05, ^##^ *p* < 0.01, ^###^ *p* < 0.001 versus matched mice at E18.5. (**H**) A schematic diagram depicting periosteal bone formation and endosteal bone resorption in femurs (cross-sections) at the days indicated. Green and red, calcein and Alizarin red S-labeled bones, respectively.

**Figure 3 ijms-22-06745-f003:**
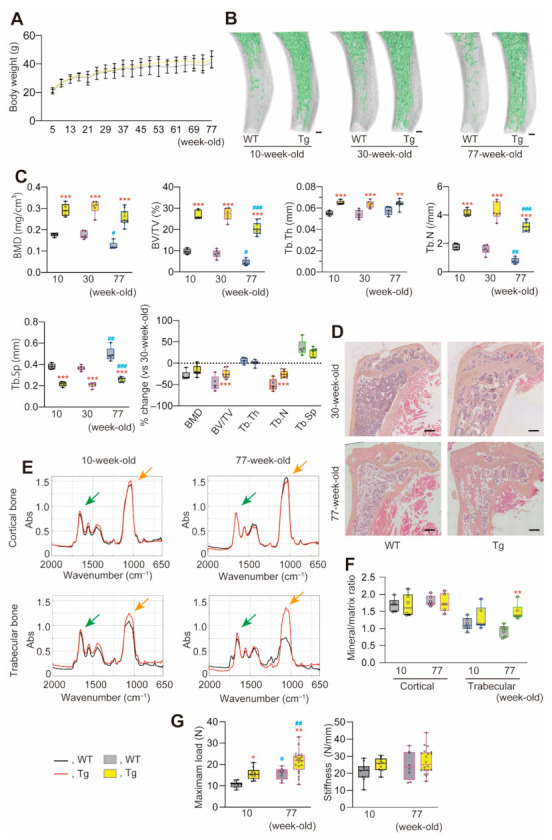
Age-dependent changes in male Tg and WT mouse growth and bone parameters. (**A**) Growth curves of male mice. n = 5–10. (**B**) Representative μCT images of proximal tibiae (longitudinal axis). Green, trabecular bone; gray, cortical bone. (**C**) Trabecular bone CT parameters in images shown in (**B**). Tb.Th, trabcular thickness; Tb.N, trabecular number; Tb.S, trabecular separation. ROI, metaphyseal region with 7 m thickness, based on the position 1 m away from the distal tibial growth plate. n = 4–9. ** *p* < 0.01, *** *p* < 0.001 versus matched WT; ^#^ *p* < 0.05, ^##^ *p* < 0.01, ^###^ *p* < 0.001 versus matched 10-week-old mice. (**D**) Representative images of proximal tibiae stained with H-E. Scale bars, 250 μm. (**E**) Representative FTIR spectra of WT and Tg tibiae. Green/orange arrows, peaks of Amide I and PO_4_^3–^ indicate collagen and mineral, respectively. (**F**) The mineral/matrix ratio was calculated from peak values in **E**. n = 6. ** *p* < 0.01 versus matched WT. (**G**) The femoral maximum load and stiffness from three-point bending tests. n = 8–20. * *p* < 0.05, ** *p* < 0.01 versus matched WT; ^#^ *p* < 0.05, ^##^ *p* < 0.01 versus matched 10-week-old mice.

**Figure 4 ijms-22-06745-f004:**
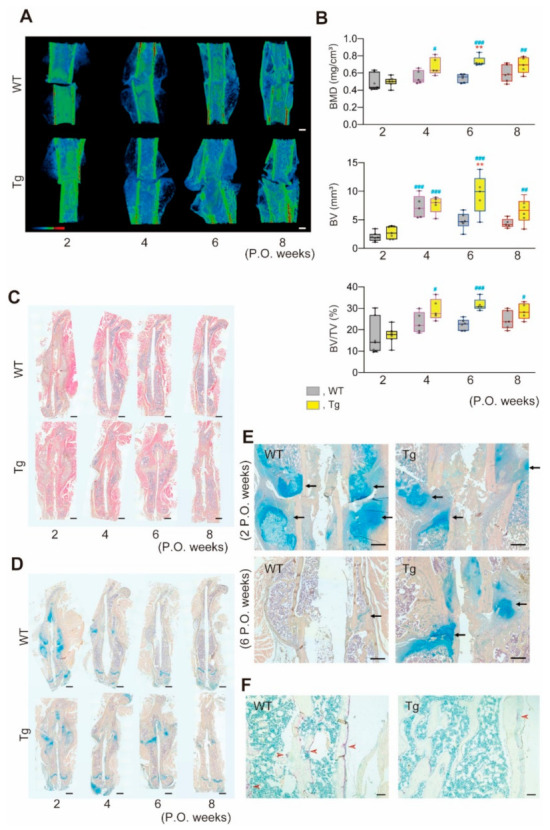
Femoral fracture healing in male Tg and WT mice. (**A**) Representative 3D μCT images of fractured femurs during postoperative (P.O.) weeks. The gradation panel responds to CT value. Scale bars, 1 mm. (**B**) Callus μCT parameters. ROI, 3 mm thickness to mesiodistal from the fracture sites. n = 7–10. ** *p* < 0.01 versus WT; ^#^ *p* < 0.05, ^##^ *p* < 0.01, ^###^ *p* < 0.001 vs. Tg at 2 P.O. weeks. (**C**–**F**) Representative histological images of longitudinal sections stained with H-E (**C**), Alcian blue and hematoxylin (**D**,**E**) and TRAP with methyl green (**F**). High magnification images of callus areas at 2 and 6 P.O. weeks (**E**) and 6 P.O. weeks (**F**). Arrows and arrowheads, cartilage and TRAP^+^ cells, respectively. Scale bars, 1.25 mm (**C**,**D**); 600 μm (**E**); 100μm (**F**).

**Table 1 ijms-22-06745-t001:** Hematology analysis (8-week-old female mice).

	WT (n = 4)	Tg (n = 4)	*p*-Value
WBC (No./μL)	1175 ± 689.81	1200 ± 355.90	0.95
RBC (10^4^/μL)	970.75 ± 53.02	903.25 ± 58.44	0.14
HGB (g/dL)	14.85 ± 0.88	14.18 ± 1.14	0.39
MCV (fL)	51.03 ± 0.98	53.15 ± 1.34	0.04
MCH (pg)	15.3 ± 0.24	15.7 ± 0.29	0.08
MCHC (%)	30.03 ± 0.36	29.53 ± 1.21	0.45
PLT (10^4^/μL)	38.58 ± 51.09	41 ± 35.30	0.94

Data are shown as mean ± S.D. WBC, white blood cell; RBC, red blood cell; HGB, hemoglobin; MCV, mean corpuscular volume; MCH, mean corpuscular hemoglobin; MCHC, mean corpuscular hemoglobin concentration; PLT, platelet.

## Data Availability

Data are contained within the article or Appendix A.

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
