# Peer review of "Overexpression of miR-125b in Osteoblasts Improves Age-Related Changes in Bone Mass and Quality through Suppression of Osteoclast Formation"

_ijms, 2021, doi:10.3390/ijms22136745_

Round 1

Reviewer 1 Report

This study aimed to investigate the effects of miR-125b overexpression under the control of OCN-promoter on skeletal morphogenesis and remodeling during development, aging, and fracture repair. Although major findings (osteopetrosis in miR-125b Tg mouse) are already published in their previous report, the current manuscript has some merits on detailed understandings of the bone formation and fracture healing process.

Major comments

In the Tg mouse, miR-125b is overexpressed under the control of human osteocalcin promoter, therefore, the bone phenotype can be traced back not only by the effect of overexpressed miR-125b but the activity of the OCN-promoter. This review recommends showing the Spatio-temporal activity of the OCN-promoter by using Ocn-lacZ of Ocn-GFP, as well as the expression pattern of the miR-125b. Referring to the corresponding data from previous studies is also encouraged to address this concern.

Is there any difference between sex? Since authors used both male and female fetuses.

Minor comments

In Fig.2H, the difference between WT and TG is hard to see. The difference is better be emphasized more clearly.

L86; Figure1C is 1B

Author Response

#1 – In the Tg mouse, miR-125b is overexpressed under the control of human osteocalcin promoter, therefore, the bone phenotype can be traced back not only by the effect of overexpressed miR-125b but the activity of the OCN-promoter. This review recommends showing the Spatio-temporal activity of the OCN-promoter by using Ocn-lacZ of Ocn-GFP, as well as the expression pattern of the miR-125b. Referring to the corresponding data from previous studies is also encouraged to address this concern.

RESPONSE – Thank you for these important comments. We have revised the relevant section of the Introduction (lines 63-66 in the revised version) by better describing the Tg model and site and level of overexpression achieved, as reported in our previous study (Minamizaki et al, Commun Biol 3, 30, 2020). We have also added a citation (new Ref. #17) in which the authors describe in detail results with the reporter mice expressing CAT under the hOC promoter (Kesterson et al, Mol Endocrinol 7, 462, 1993).

#2 – Is there any difference between sex? Since authors used both male and female fetuses.

RESPONSE – We revised the Materials and Methods section to address this important point (lines 262-265 in the revised version). For development and early postnatal analyses, fetal and newborn mice (up to P5) were used regardless of sex, since this strain exhibits no differences in femoral bone parameters at least up to P7 (Richman et al, J Bone Miner Res 16, 386, 2001). Previously, we confirmed that male and female Tg mice exhibited the same skeletal phenotypes (Manamizaki et al, Commun Biol 3, 30, 2020), but we used male mice for most (except for hematology) of the later postnatal studies reported here to avoid potential sex differences.

#3 – In Fig.2H, the difference between WT and TG is hard to see. The difference is better be emphasized more clearly.

RESPONSE – As the reviewer suggested, Fig. 2H has been modified to show more clearly that calcein-labeled unresorbed bone remains in Tg mice, with a consequent increase in bone mass; the corresponding figure legend has also been modified to improve clarity.

#4 – L86; Figure1C is 1B

RESPONSE – We apologize for the mistake which has been corrected.

Reviewer 2 Report

Abstract

Please refer to the potential clinical impact that miR-125b may have.

Introduction

The connection to fracture healing is not deduced sufficiently. The first part of the introduction gives a solid overview of the field, but I am missing a development of your hypothesis.

Also, please give a brief description of your used Tg model.

Results

"possibly via decreased osteoclastic bone resorption": Did you measure Oc.N?

Fig. 1: Where is the quantification?

Fig. 2F: The upper part of the bone seems washed-out, please provide a proper high-res image. H: How was the model created?

Fig. 3: "but the decrease in BV/TV and Tb.N was less in Tg compared to WT mice": Do you have a statistical validation for that statement?

"Leading us to quantify standard blood cell types": Where were these assessed from? Peripheral blood does not necessarily correlate to bone marrow populations.

Were the detected differences in micro-CT also visible biomechanically? I.e. are these mechanically evident differences?

It is unexpected that bone stiffness is increased with aging, do you have an explanation for that? Why is there no 30-week value?

Did you stabilize the bones in order to heal? If so, how? If not, then state that the unstabilized femoral fracture is more an exception than the rule.

Did you perform micro-CT in vivo? How and how often? Please report that, even in the description of the model. How did you make sure the frequency of radiation (by ct) did not interfere with the healing process? As radiation is known to interfere with physiological bone healing.

Please give a biomechanical characterization of the "healed" bone to underline your statement that Td does delay the healing process like the result itself.

Discussion

Why do you state that further research in chondrocytes is needed, but show not a single chondrocyte-specific effect in your results? Please delete the respective part.

line 224-226: A regular blood count cannot give a proper description of the underlying bone marrow.

Please give a proper explanation of why neither strength nor stiffness decline with aging.

line 248-263: This paragraph does not add anyhing; how can you be sure that bisphosphonates interfere with your miR-125b model?

Please give a descrition of the mouse model, not just a reference to it since it is essential for the whole study.

uCT: When was it done? How long did it take? Were the mice anesthesized for it?

The three-point bending test is less reliable compared to a torsion test. Either do the torsion test - or discuss the limitations of the bending test.

Assessment of fractured femurs on week 2,4,6 and 8: What did radiation do? As example, please see: PMID: 31768003.

Statistics: Comparison via students t test: Did you assess parametricity?

Where do you show mineral apposition rate? You report the mineral/matrix ratio, right?
Where are the osteoclast numbers?

You do not show the miR-125b overexpression in any cell type, please do so. At least in osteoblasts.

Author Response

#1 – Abstract; Please refer to the potential clinical impact that miR-125b may have.

RESPONSE – Thank you for the suggestion. We had presented a short description at the end of the original version of the Abstract, but have modified the ending to better make the point around potential clinical impact. To this end, we also revised the end of the Introduction and the Discussion.

#2 – Introduction; The connection to fracture healing is not deduced sufficiently. The first part of the introduction gives a solid overview of the field, but I am missing a development of your hypothesis.

RESPONSE – We apologize for the lack of clarity in explaining our aim. Our previous study showed a marked increase in bone mass due to the suppression of osteoclastogenesis by miR-125b encapsulated in matrix vesicles budding from osteoblasts. These findings open the possibility of targeting this process for clinical application in patients with bone loss. However, because some antiresorptive agents are known to increase impaired/delayed fracture healing, we wanted to investigate the integrity of fracture healing in our Tg mouse model. We have now revised the Introduction to explain this point (lines 76-77 in the revised version).

#3 – Also, please give a brief description of your used Tg model.

RESPONSE – As the reviewer suggested, a better description of the model (miR-125b selectively overexpressed in osteoblasts of Tg mice) has been added to the Introduction section (lines 61-69 in the revised version).

#4 – Results; "possibly via decreased osteoclastic bone resorption": Did you measure Oc.N? Fig. 1: Where is the quantification?

RESPONSE – We apologize for the lack of clarity in presenting our results. We have revised the wording to describe and clarify the quantitative analysis (lines 93-95 in the revised version) shown in Fig. 2E.

#5 – Fig. 2F: The upper part of the bone seems washed-out, please provide a proper high-res image. How was the model created?

RESPONSE – We have analyzed the overall femur including both epiphysial ends. However, because the femoral epiphyses at early stages of development are occupied by cartilage (umnineralized), their details cannot be visualized by microCT and the “washed out” appearance reflects this.

#6 – Fig. 3: "but the decrease in BV/TV and Tb.N was less in Tg compared to WT mice": Do you have a statistical validation for that statement?

RESPONSE – As shown in Fig. 3c (% change vs 30-week-old) and the corresponding legend, the decreases in BV/TV and Tb.N in Tg mice were significantly lower than those in WT mice.

#7 – "Leading us to quantify standard blood cell types": Where were these assessed from? Peripheral blood does not necessarily correlate to bone marrow populations.

RESPONSE – We apologize that we have not made our points clearly enough in the original version of the manuscript. As the reviewer pointed out, the hematology analysis was not done directly to examine bone marrow populations. However, a decreased marrow cavity may lead to anemia and bleeding, infections and hepatosplenomegaly due to increased extramedullary hematopoiesis (i.e., osteopetrosis, Penna et al, Dis Model Mech 14, 2021). Although we did not detect abnormal anatomy suggesting hepatosplenomegaly, we nevertheless thought it was important to address the point further by performing the hematology analysis. We have now modified the text to make the purpose of the analysis clear (lines 135-136 in the revised version), and have also added to the Discussion section (lines 226-227 in the revised version) the following sentence “However, further investigation is needed to explore the possibility of extramedullary hematopoiesis in Tg mice.”

#8 – Were the detected differences in micro-CT also visible biomechanically? I.e. are these mechanically evident differences?

It is unexpected that bone stiffness is increased with aging, do you have an explanation for that? Why is there no 30-week value?

RESPONSE – We thank the reviewer for these valuable comments. Based on the microCT data, we predicted that the Tg bones would be mechanically weakened, but the 3-point bending test did not support this.

We have data of 30-week-old Tg and WT mice, which showed that there were no significant differences compared with those of 10-week-old mice. Therefore, because they were not informative in understanding the differences between the two genotypes, we chose not to show the 30-week-old data. Ferguson et al (Bone 33, 387, 2003) reported that bone strength and stiffness in C57BL/6J mice, the background strain used here, peaked at 52-weeks of age and thereafter began to decrease (Ref #27 in the original version of our manuscript). Thus, although we may have analyzed an age marginally (two weeks) too young to detect peak bone strength, our data show obvious and statistically significant differences between Tg and WT mice. We have modified the descriptions to make these points more clearly (from line 238 in the revised version).

#9 – Did you stabilize the bones in order to heal? If so, how? If not, then state that the unstabilized femoral fracture is more an exception than the rule.

RESPONSE – Again, we apologize for the lack of clarity. We used the currently accepted method by which fractured femurs were fixed using a pin intramedullary, as described in the Materials and Methods section and Supplementary Fig.1.

#10 – Did you perform micro-CT in vivo? How and how often? Please report that, even in the description of the model. How did you make sure the frequency of radiation (by ct) did not interfere with the healing process? As radiation is known to interfere with physiological bone healing.

RESPONSE – We agree with the reviewer’s comments and apologize that we did not describe the methods clearly. In our studies, microCT was not done in vivo but on isolated bone preparations. We have now added this to the Materials and Methods (line 286 in the revised version).

#11 – Please give a biomechanical characterization of the "healed" bone to underline your statement that Td does delay the healing process like the result itself.

RESPONSE – We agree with the reviewer’s comment. In this manuscript, we have shown histological findings but not biomechanical characterization of the repairing fracture. We have therefore replaced “healing” with “callus resorption.”

#12 – Discussion; Why do you state that further research in chondrocytes is needed, but show not a single chondrocyte-specific effect in your results? Please delete the respective part.

RESPONSE – We agree with the reviewer’s comment and suggestion, and have deleted the sentence.

#13 – line 224-226: A regular blood count cannot give a proper description of the underlying bone marrow.

RESPONSE – We agree; please see our response to #7.

#14 – Please give a proper explanation of why neither strength nor stiffness decline with aging.

RESPONSE – Please see our response to #8.

#15 – line 248-263: This paragraph does not add anyhing; how can you be sure that bisphosphonates interfere with your miR-125b model?

RESPONSE – We agree with the reviewer’s comment and have deleted the sentence.

#16 – Please give a description of the mouse model, not just a reference to it since it is essential for the whole study.

RESPONSE – Please see our response to #3.

#17 – uCT: When was it done? How long did it take? Were the mice anesthesized for it?

RESPONSE – As noted in our response to #10, microCT was done on isolated bone preparations, not mice.

#18 – The three-point bending test is less reliable compared to a torsion test. Either do the torsion test - or discuss the limitations of the bending test.

RESPONSE – In addition to our response to #8, we have also added a description to the Results section (lines 147-148 in the revised version).

#19 – Assessment of fractured femurs on week 2,4,6 and 8: What did radiation do? As example, please see: PMID: 31768003.

RESPONSE – Please see our responses to #10 and #17.

#20 – Statistics: Comparison via students t test: Did you assess parametricity?

RESPONSE – Thank you for catching this error in describing our methods; yes, we tested parametricity and corrected the methods to state use of Welch’s t test.

#21 – Where do you show mineral apposition rate? You report the mineral/matrix ratio, right?
Where are the osteoclast numbers?

RESPONSE – We apologize for misstating what was done, and have corrected “mineral apposition rate” to assessment of the presence of calcein-labeled bone in the Abstract and Conclusion sections.

#22 – You do not show the miR-125b overexpression in any cell type, please do so. At least in osteoblasts.

RESPONSE – We apologize for the lack of clarity in presenting our results. We have added a brief description of this issue to the Introduction section, and our previous study in which we quantified miR-125 overexpression has been cited (from line 63 in the revised version).

Round 2

Reviewer 2 Report

Fig. 3: You state that the "decrease in BV/TV and Tb.N was less in Tg compared to WT mice". Indeed, the statistical verification of a decrease from 30 to 77 shows significance in both WT and Tg. But did you directly compare the (relative) decrease from 30 to 77 weeks in both and compare whether Tg "lost" less than WT? This could be done by a pairwise comparison of the (relative) change from 30 to 77. Otherwise, the statement still cannot be held.

"We thank the reviewer for these valuable comments. Based on the microCT data, we predicted that the Tg bones would be mechanically weakened, but the 3-point bending test did not support this." I think that a biomechanical analysis of 30-week old mice would further strengthen your results. Although the effect may be not significant, I strongly support demonstration this (negative) result. Moreover, the missing biomechanical correlation may be relevant for future studies and other groups.

Please add "ex vivo" in the the micro-CT description.